# School Climate and Responsibility as Predictors of Antisocial and Prosocial Behaviors and Violence: A Study towards Self-Determination Theory

**DOI:** 10.3390/bs11030036

**Published:** 2021-03-17

**Authors:** David Manzano-Sánchez, Alberto Gómez-Mármol, Alfonso Valero-Valenzuela, José Francisco Jiménez-Parra

**Affiliations:** 1Faculty of Sport Sciences, University of Murcia, 30720 Santiago de la Ribera, Spain; david.manzano@um.es (D.M.-S.); josefrancisco.jimenezp@um.es (J.F.J.-P.); 2Faculty of Education, University of Murcia, 30100 Murcia, Spain; alberto.gomez1@um.es

**Keywords:** psychological basic needs, autonomous motivation, education, school, teenagers, children

## Abstract

Self-determination theory and Vallerand’s hierarchical model have been studied taking into account different types of social factors that can result in different consequences. The purpose of this work was to see if responsibility and social climate could predict antisocial and prosocial behavior and violence. For this, 429 students (M = 11.46, SD = 1.92) participated in the study, answering a questionnaire with five variables: school climate, responsibility, motivation, satisfaction of psychological needs, prosocial and antisocial behavior, and violence. The main results indicated that most variables correlated positively and directly, except in the case of antisocial behavior and violence. On the other hand, a prediction model (X2 = 584.145 (98); RMSEA = 0.104 [90% CI = 0.096, 0.112]; TLI = 0.849; CFI = 0.894) showed that responsibility and school climate can predict basic psychological needs, and that these needs can improve autonomous motivation, which, in turn, could positively predict on improving prosocial behavior and reducing antisocial behavior and violence. In conclusion, school climate and responsibility can encourage the development of positive consequences in the classroom, specifically in terms of prosocial behavior and the reduction of violence and antisocial behavior.

## 1. Introduction

Pre-adolescence and adolescence are considered to be stages of life in which externalizing problems, such as antisocial behavior (understood as behavior that intentionally causes harm or damage to another person [1]) or violence, and internalizing problems, such as shyness or social anxiety [2], increase significantly, with violence being considered one of the most important factors that the education system must address at the international level [3]. This is due to its close relationship with antisocial behavior and its association with negative consequences on mental health and personal, social, and school adjustment [4].

Given this situation, recent studies suggest the need to pay greater attention to the education in values that young people receive, as an element aimed at reducing school violence and social conflict [3], and, in turn, improving prosocial behavior, understood as those forms of voluntary behavior that are aimed at creating positive interpersonal relationships and maintaining personal and social well-being [5].

On the other hand, school social climate is understood as the perception of students and teachers regarding the quality of their classroom experiences [6], normally characterized by the establishment of satisfactory socio-affective interactions between them. This variable contributes to the formation of an adequate classroom environment within the teaching–learning process [7], with the intention of achieving greater social and academic performance [8], as well as reducing bullying and violence in schools [9].

Thus, the field of education is an ideal place for the processes of character formation and the development of social skills, being a field of knowledge that is attracting ever-increasing research developed from different disciplines, such as psychology and pedagogy [10]. In this respect, several recent research papers [11,12,13] have already stated that a low level of social skills in young people and adolescents may be the trigger for violent episodes in school contexts. Similarly, Cunha et al. [14] reported the association between the prevalence of school violence (which is rising internationally, according to Rocha et al. [15]) and the creation of a negative school climate that, according to Carbonero et al. [16], is characterized by low levels of motivation, which are linked with a greater deficit in basic psychological needs [17]. To prevent the development of a negative school climate, Lenz et al. [18] emphasized the importance of psychological well-being, academic achievements, and good school coexistence.

Thus, in the school context, the promotion of values through personal and social responsibility is a key element in improving prosocial behavior [3,19], reducing student violence [7,20], and improving classroom climate [7].

Courel-Ibañez et al. [21] suggest the establishment of structured programs to promote personal and social responsibility as a trigger for positive behavioral, affective, and emotional consequences. One pedagogical model that attaches “best consequences” with regard to value development is the personal and social responsibility model (TPSR), designed by Donald Hellison [22]. This model divides the formation of personality into four areas: on one hand, engagement/effort and autonomy as elements of personal responsibility, and, on the other hand, respect for others and help/leadership as elements of social responsibility. In addition to this framework there is a fifth area, which aims to extrapolate what has been learned outside of the context of school, that is, in everyday life settings. These four areas can be classified as social skills [23]. Different empirical studies have analyzed the model’s effect both in the out-of-school environment [24] and in the school environment [7], showing its effectiveness in reducing school violence and improving personal and social responsibility, motivation, basic psychological needs, school climate, and prosocial behavior. Literature reviews by Pozo et al. [25] and Sánchez-Alcaraz et al. [26] of major international databases conclude that the most common positive effects reported after the implementation of TPSR are the improvement of social behavior, interpersonal relationships, self-control, and self-efficacy, and values such as respect, autonomy, leadership, and helping others.

In order to understand motivational processes, this research is framed under the theoretical construct proposed by Deci and Ryan’s self-determination theory [27,28], which understands motivation as a continuum ranging from demotivation to intrinsic motivation [28]. In addition, it analyses its origin and consequences at the cognitive, behavioral, and affective levels in the person [29]. This theoretical construct argues that basic psychological needs are composed of autonomy, competence, and social relatedness. The satisfaction of these needs and the origin of the level of motivational regulation will explain certain behavior or psychosocial consequences [30]. In this sense, several studies have confirmed that higher levels of basic psychological need satisfaction are found in those participants with a self-determined motivation [31,32,33], and they place the positive consequences at the cognitive, affective, and behavioral levels (consequent variables). It has been shown that students who have greater satisfaction with BPN develop a more self-determined motivation [34], and, in turn, the motivational processes developed by the students act as determining elements in behavior developed during the classes [35].

Within self-determination theory [28], Vallerand’s hierarchical model [29,30] establishes that the impact of social factors, such as responsibility and school social climate, are mediated by the basic psychological needs of autonomy, competence, and relatedness, whose satisfaction is considered fundamental to promote more self-determined motivational states [36] and generate positive consequences at the cognitive, affective, and behavioral levels [37], such as prosocial behavior. The sequence established by Vallerand [30] has been studied by several authors with different types of social factors and consequences [38,39]. However, no study has analyzed the model taking into account responsibility as a social factor and prosocial/antisocial behavior and violence as a consequence of this theoretical construct.

Only the study by Menéndez and Fernández-Río [32] took into account responsibility as a social factor and the goal of friendship-approach as a consequence, obtaining results that reflect the capacity of responsibility to predict the satisfaction of BPN, intrinsic motivation, and the goal of friendship-approach. Different works in this line [32,40,41] highlight the importance of analyzing social domains in order to have a better understanding of the motivation of adolescent students.

Despite the fact that in the scientific literature, there is an increasing amount of work that analyzes these variables separately, or even links them with other third variables, up to now, and to the best of our knowledge, this is the first research that aims to determine not only if there is a correlation between all of them, but also, the predictive capability of responsibility and social climate with regard to BPN, autonomous motivation, prosocial/antisocial behavior, and violence. The main reason for carrying out this study was that the establishment of learning situations based on responsibility and the promotion of an adequate social climate in the school environment could lead to a satisfaction of students’ basic psychological needs and an increase in their autonomous motivation, which in turn would predict an improvement in prosocial behavior and a reduction in violence.

## 2. Materials and Methods

### 2.1. Participants

A total of 429 primary school (fifth and sixth grade) and secondary school (from first to fourth grade) students from three different schools, with similar sociodemographic characteristics, participated in this study (227 men, 52.9%; 202 women, 47.1%). Their average age was 11.46 (SD = 1.92). According to age, 294 were from primary school (68.5%) and 135 from secondary school (31.5%). They were selected based on accessibility and convenience.

### 2.2. Instruments

Sociodemographic variables (gender and age) were answered by the students, as well as a multiple questionnaire with these scales:(1)The Personal and Social Responsibility Questionnaire (PSRQ) was used to measure personal and social responsibility levels. It was adapted to the school context by Li et al. [42], and for Spanish by Escartí et al. [43], and validated in a sample of 9-to-15-year-olds. This scale consists of 14 items, seven to assess social responsibility (e.g., “I help others”) and seven for personal responsibility (e.g., “I set goals”). The answers were provided on a Likert-type scale ranging from 1 (totally disagree) to 6 (totally agree). Reliability in the test was 0.82 for social responsibility and 0.82 for personal responsibility. Total responsibility (the mean of social and personal responsibility) had a reliability of 0.89.(2)A questionnaire to assess social school climate (CECSCE) was used to evaluate the climate perceived by the students with regard to their class, teacher, and school. It was designed by Trianes et al. [44] and validated in a sample of 12-to-14-year-olds. The questionnaire consists of two subscales called “center climate” (with questions about the climate in the school and in the class, e.g., “Students are really willing to learn”), made up of eight items, and “teaching climate” (e.g., “Teachers of this school are friendly to students”), composed of six items. A five-point Likert-type scale was used, ranging from 1 (totally disagree) to 5 (totally agree). The internal consistency analysis yielded a value of 0.85 for center climate and 0.69 for teaching climate. Both scales make up the school climate (general scale value), which had a reliability of 0.81.(3)The Psychological Need Satisfaction in Exercise (PNSE) was used to measure the satisfaction of the needs for social competence, autonomy, and relatedness. The scale was adapted for Spanish and to the education context by Moreno-Murcia et al. [45], and validated in a sample of 12-to-16-year-olds. This scale consists of 18 items, six to evaluate each need: competence (e.g., “I am confident to perform the most challenging tasks”), autonomy (e.g., “I believe I can make decisions during my classes”), and relatedness with others (e.g., “I feel attached to my classmates because they accept me as I am”). These were preceded by the sentence “During my class…”, and the answers were provided on a Likert-type scale ranging from 1 (false) to 6 (true). Reliability in the pre-test was 0.70 for autonomy, 0.76 for competence, and 0.71 for relatedness. Moreover, the psychological mediator index (PMI) was applied to evaluate the three variables jointly, yielding an internal consistency of 0.84.(4)The Motivation Toward Education Scale (in French, EME) was used to measure motivation from the most self-determined types to the most external causes and amotivation. The Spanish version of the Échelle de Motivation en Éducation [46] validated by Nuñez et al. [47] was used. The questionnaire passed a reliability test in order to check the understanding of the student sample in the same way as the others. This study used the denominated “autonomous motivation” as recommended by Sánchez-Oliva et al. [48], composed of 4 scales: intrinsic motivation to knowledge (e.g., “because I feel pleasure and satisfaction when I learn new things”), to accomplishment (e.g., “for the pleasure I feel when I improve my academic performance”), to experience sensations (e.g., “because reading about topics I find interesting stimulates me”), and identified regulation (e.g., “because it will allow me to access to the job market in my preferred field”). Autonomous motivation is composed of 16 items (four items for each scale) preceded by the sentence “I go to school/high school because…”, with a seven-point Likert-type scale, from 1 (totally disagree) to 7 (totally agree). The reliability values were 0.78 (intrinsic motivation to know), 0.80 (intrinsic motivation for accomplishment), 0.74 (intrinsic motivation to experience), and 0.70 (identified regulation). Finally, the reliability of autonomous motivation was 0.79.(5)The Scholar Violence Questionnaire (CUVE) from Álvarez et al. [49] is divided into a version for secondary school, with 8 subscales, and one for primary, with 7 subscales. It was adapted to Spanish and to the context of primary and secondary school by Álvarez et al. [50]. In the case of secondary school, the subscale of “violence through information and communication technologies” is included (e.g., “students publish on the internet offensive photos or videos of colleagues”); it was deleted in this study to check the same scales for primary and secondary students. The other sub-scales that make up the questionnaire and their internal consistency were as follows: verbal violence towards students (e.g., “students speak badly about each other”, four items, α = 0.73), verbal violence towards teachers (e.g., “students speak with bad manners to teachers”, four items, α = 0.77), direct physical violence between students (e.g., “students engage in fights on school grounds”, five items, α = 0.68), indirect physical violence by students (e.g., “ students steal things from teachers”, four items, α = 0.77), social exclusion (e.g., “ certain students are discriminated against by their classmates”, seven items, α = 0.82), disruption in the classroom (e.g., “ there are students who neither work nor let others work “, three items, α = 0.61), and teacher violence towards students (e.g., “teachers do not listen to their students”, seven items, α = 0.83). The total internal consistency of the questionnaire was 0.93 for primary and 0.91 for secondary students. The responses are collected in a Likert-type scale whose scoring ranges from 1 (totally disagree) to 5 (totally agree).(6)The Teenager Inventory of Social Skills (TISS) from Inderbitzen and Foster [51] was used to evaluate prosocial and antisocial behavior, and was adapted to Spanish by Inglés et al. [52]. The questionnaire is made up of two subscales: pro-social values (21 items), including positive social behavior such as cooperation, community participation, altruism, and the ability to express feelings (e.g., “I offer help to my classmates to do their homework”); and antisocial values (19 items), including aggression, low self-esteem, social anxiety, presumption, and insolence (e.g., “I forget to return things that others have lent me”). It uses a five-point Likert-type scale, from 1 (“it does not describe anything about me”) to 6 (“it fully describes me”). The internal consistency values were 0.89 for the prosocial values scale and 0.87 for the antisocial values scale.

### 2.3. Procedures

Before completing the questionnaire, the research team contacted the different centers. After that, the participants were given an information sheet and were asked to sign an informed consent form. The students answered a questionnaire in a 35 min session in a quiet environment. First, students watched a PowerPoint presentation about how to complete the questionnaires. After that, the teacher read the questions in order to ensure they were understood. The teacher and one of the researchers stayed with them all the time to solve possible doubts. The participants were requested to provide truthful answers. Participants were informed of the purpose of the research and were told that it was voluntary and confidential.

This study previously received the approval of the Ethics Committee of the University of Murcia (1685/2017). All participants were dealt with following the ethical guidelines concerning consent, confidentiality, and anonymity of the answers. In addition, an informed consent was made by the parents and the directors of the schools.

### 2.4. Statistical Analysis

Means, standard deviation, and bivariate correlations were analyzed for all variables under analysis. A two-step maximum likelihood (ML) approach suggested by Kline [53] in AMOS 23.0 (SPSS Inc., Chicago, IL, USA) was performed. Firstly, confirmatory factor analysis (CFA) was performed to analyze the psychometric properties of the proposed model. A composite reliability via Raykov [54] formula was performed to assess internal consistency, taking 0.70 as the cut-off value [55], while the average variance extracted (AVE) was estimated for analyzed convergent validity [55].

Discriminant validity was established when the correlation coefficients were lower than the AVE for each construct exceeding the squared correlations between that construct and any other construct [56]. Secondly, a structural equation model (SEM) was performed to test proposed relationships among different constructs. For CFA and SEM, the following absolute and incremental indices were used for analysis: comparative fit index (CFI), normalized fit index (NFI) and root mean square error of approximation (RMSEA) with its confidence interval (CI: 90%). For these indices, scores of CFI and NFI > 0.90 SRMR and RMSEA < 0.08 were considered as acceptable, following several recommendations [55,57,58].

## 3. Results

### 3.1. Descriptive and Sociodemographic Variables Analysis

Descriptive values are in Table 1. The asymmetry and kurtosis values were for all variables <3 and <10, respectively, and the value of α was >0.70, except for teacher climate, although this was very close (α = 0.69).

Table 2 shows the correlations between age and the variables of the model. Specifically, age was correlated with responsibility, school climate, PMI, and autonomous motivation (negative); and with antisocial behavior and violence (positive).

Table 3 shows the differences according to gender. The differences were of *p* < 0.01 for responsibility (F = 10.177), school climate (F = 9.734), prosocial behaviors (F = 10.979), and antisocial behaviors (F = 29.662), with higher values in girls except for antisocial behaviors. A value of *p* < 0.05 was found for violence in favor of boys (F = 5.254). 

### 3.2. Measurement Model

The number of latent variables per factor was reduced in order to conduct the analysis of the measurement model and then test the structural equation model (SEM). To do so, the items for antisocial, prosocial, and violence were grouped into pairs [37]. The model was, therefore, identified, since every latent variable was measured by at least two indicators [59]. Mardia’s coefficient (39.94) was used to check factors’ multivariate normality, it being lower than 70 [60]. In addition, the multicollinearity assumption was met, since all bivariate correlations between variables were below 0.85. The errors among the endogenous variables were independent, since they were not correlated with other variables. The maximum likelihood estimation method was applied.

Table 4 shows the bivariate correlations among variables. The majority of the variables had a significant correlation among them. For instance, responsibility and school climate is positively and significantly associated with PMI, while autonomous motivation and prosocial behavior are negatively and significantly associated with antisocial behavior and violence. Finally, all constructs present adjusted values of composite reliability, all greater than 0.70 [39].

The test of the measurement model included responsibility, school climate, PMI, autonomous motivation, prosocial behavior, antisocial behavior, and violence. Results show a good fit with the data (X2 = 393.405 (98); RMSEA = 0.084 [90% CI = 0.075, 0.093]; SRMR = 0.699; TLI = 0.908; CFI = 0.928; NFI = 0.901). Additionally, the measurement model revealed no problems of convergent or discriminant validity, since the average variance extracted (AVE) followed the recommendations by Hair et al. [55] and Fornell and Larcker [56], and the square correlations among all constructs were less than the AVE of each factor [56].

### 3.3. Structural Model

The structural model (Figure 1) is close to the good fit data values specified in the statistical analysis section (X2 = 584.145 (98); RMSEA = 0.104 [90% CI = 0.096, 0.112]; TLI = 0.849; CFI = 0.894). All regression weights show statistical differences of *p* < 0.01. In standardized direct effect (Figure 1), significant associations were observed among all con-structs. Specifically, a positive correlation between school climate and responsibility (β = 0.84), a direct association between responsibility and PMI (β = 0.37) and school climate and PMI (β = 0.63), as well as between PMI and autonomous motivation (β = 0.79). Regarding the final prediction, PMI was significant, showing a positive association with pro-social behavior (β = 0.27), and negative associations with antisocial behavior (β = −0.25) and violence (β = −0.47). School climate had a significant positive effect on prosocial behaviors (β = 0.208, *p* < 0.001), and a negative effect on violence (β = −0.496, *p* < 0.001). On the other hand, responsibility had a significant positive effect on prosocial behavior (β = 0.334, *p* < 0.001), and negative effects on violence (β = −0.239, *p* < 0.001) and antisocial behavior (β = −0.145, *p* = 0.005). Finally, it is supported that PMI and autonomous motivation mediate the relationship between school climate and antisocial behavior (β = 0.146, *p* = 0.027), responsibility and antisocial behavior (β = 0.133, *p* = 0.022), and responsibility and prosocial behavior (β = 0.110, *p* = 0.006).

## 4. Discussion

The main objective of this study was to analyze the predictive capacity of responsibility and the social climate on BPN, autonomous motivation, prosocial/antisocial behavior, and violence. The structural model is sustained on acceptable values except CFI, which was very slightly lower (0.894) than the recommendations (CFI > 0.90) [42]. Nevertheless, this value (0.894) was considered valid by Requena [61], because, as happened in this study, all the factors’ loads were over 0.30 (by far), which is, according to Méndez and Rondón [62], the strictest statistic criteria for confirmatory factorial analysis. The results of the structural model reflect that the regression shows statistical differences among all the constructs. Specifically, positive and significant correlations are observed between school climate and responsibility, as well as direct associations between responsibility and PMI, school climate and PMI, and between PMI and autonomous motivation. The research aim was to study the relationship between, on the one hand, responsibility and social climate, and on the other, satisfaction of students’ basic psychological needs, autonomous motivation, prosocial and antisocial behavior, and violence. The hypothesis and the model to test was that both of them (responsibility and school climate) could predict the satisfaction of students’ basic psychological needs and their autonomous motivation, which in turn would predict prosocial and antisocial behavior and violence. 

Therefore, the results obtained practically confirm the hypothesis raised. As has been previously indicated, although there is a growing amount of work in the scientific literature that analyzes these variables separately or even links them to other third variables, there are not yet, at least to our knowledge, any that have analyzed the correlation be-tween all the variables under study. Only one work [7] reflects results very similar to those of the present study, finding a positive and significant association of responsibility with school social climate, basic psychological needs (autonomy, competence, and relationship), PMI, self-determination index, and prosocial behavior, as well as a negative association of responsibility with demotivation, antisocial behavior, and violence.

On the other hand, although there are no studies that have analyzed the predictive capacity of responsibility and social climate on the rest of the variables, in which prosocial/antisocial behavior and violence are located as a final consequence of Vallerand’s hierarchical model [29], a study was found [16] in which the same sequence with the same social factor was studied, but with different consequences. In this study [32], they analyzed a theoretical model under the self-determination theory, in which responsibility was taken into account as a social factor and the goal of friendship-approach as a consequence in Vallerand’s hierarchical model [29]. The results revealed that responsibility, BPN, and intrinsic motivation significantly predicted the goal of friendship-approach. The bivariate correlations carried out also showed the significant and positive correlations of the variables. Thus, the results are related to those obtained in the present study, where responsibility, BPN, and motivation were positively and significantly correlated.

Other studies in this line considered the same sequence, but with different factors and consequences [38,39,63]—specifically, the research of Moreno-Murcia et al. [39], where they analyzed the predictive capacity of social goals, BPN, and intrinsic motivation on effort. As in the present study, significant correlations were found between the variables of responsibility, BPN, and intrinsic motivation, which could be an appropriate sequence to predict the consequences under investigation. Following Vallerand’s hierarchical model, it can be said that responsibility is a social factor to be taken into account due not only to its capacity to predict BPN and most self-determined motivation, but also because of the high correlation existing between all these variables. Furthermore, it is observed that this theoretical sequence is capable of predicting different final consequences such as effort [39], the approximation-friendship goal [32], and prosocial/antisocial behavior and violence, analyzed in the present study.

From a practical and pedagogical point of view, Menéndez and Fernández-Río [32] suggest the importance of teachers applying methodologies in the educational context aimed at fostering the development of these variables through the use of pedagogical models such as cooperative learning [64], sports education [65], or teaching personal and social responsibility (TPSR) [66]. The latter has been applied in different educational contexts, such as in extracurricular activities [67], in the school environment in the area of physical education [68], and in the rest of the curricular subjects [7], demonstrating the effectiveness of its implementation to improve responsibility, autonomy, motivation, and school social climate [6,69]. In this way, the use of pedagogical approaches in the educational context can allow the necessary conditions to be reached to promote responsibility, BPN, and self-determined motivation of students in the classroom [32], creating a school social climate that favors the development of prosocial behavior among students, while decreasing antisocial and violent behavior. In fact, the study by Courel-Ibáñez et al. [21] concluded that improved development of personal and social responsibility in adolescents will contribute significantly to a reduction in violent behavior.

Thus, the TPSR is positioned as an appropriate pedagogical model to promote education in values, reduce school violence [3,70,71], and promote student autonomy [70], since new teaching approaches focused on student interests allow for greater satisfaction of BPN and greater intrinsic motivation values, resulting in better social behavior among students. In the present study, it is observed how both variables positively and significantly predicted students’ prosocial behavior. Valero-Valenzuela et al. [72] conclude that the use of teaching models that promote the support of autonomy through the assignment of responsibilities produce multiple benefits, among which the satisfaction of students’ BPN, primarily that of autonomy, stands out. This need shows a negative correlation with variables to be eradicated in the educational context, such as violent or disruptive behavior [7,73].

The present study also presents a series of limitations that should be considered for future research. Only three centers with similar socioeconomic characteristics participated in the research. Therefore, new research should be carried out in centers of diverse socioeconomic origin in order to obtain more valid results. This kind of sampling has been intentional due to accessibility. Future work that addresses this issue should be carried out using sampling with greater methodological validity, such as random sampling. Finally, the type of methodological design, of a transversal and correlational nature, prevents any type of explanation of a causal nature. Longitudinal studies and experimental and/or quasi-experimental designs should be carried out to check the sequence proposed in this study. Besides these methodological concerns, other prospective research could analyze the role that sportsmanship has in this structural model or how it is changed if the sample is made up of university students.

## 5. Conclusions

This study reflects the importance of developing school climate and responsibility in the educational context, due to its ability to promote and predict the satisfaction of BPN. It also reflects that satisfaction of these needs may predict an improvement in autonomous motivation, which, in turn, could predict an improvement in prosocial behavior and a reduction in student antisocial behavior and violence. The results of this research show the need to promote these types of variables, from the teachers’ instruction, and in educational centers through the use of methodological approaches oriented towards the students’ motivational processes, which in this case could be the pedagogical models, and more particularly the TPSR.

On the other hand, the increase in prosocial behavior and the reduction of antisocial behavior and violence were associated with a more autonomous motivation. The inclusion of prosocial/antisocial behavior and violence as variables within Vallerand’s hierarchical model is a novel element of this research that can help other researchers to analyze motivation in the educational field from a social point of view.

For this reason, the improvement of school climate and responsibility could help centers to increase prosocial behavior and decrease antisocial behavior and violence. In addition, the positive and significant connection between these variables could be considered a reference point in the theoretical framework of motivation to analyze social factors.

## Figures and Tables

**Figure 1 behavsci-11-00036-f001:**
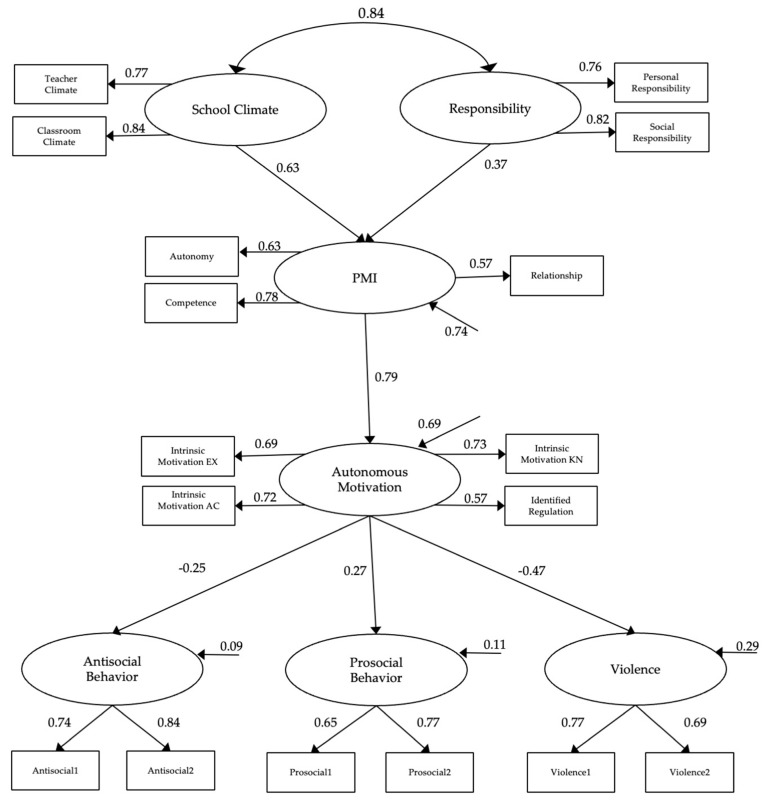
Model with relationships between school climate, responsibility, psychological mediator index (PMI), autonomous motivation, prosocial and antisocial behavior, and violence.

**Table 1 behavsci-11-00036-t001:** Descriptive values.

Variables	M	SD	Range	α	Asymmetry	Kurtosis
Intrinsic motivation to knowledge	5.06	1.63	1–7	0.78	−0.839	−0.222
Intrinsic motivation to accomplishment	5.59	1.26	1–7	0.80	−1.082	1.001
Intrinsic motivation to experience	4.91	1.36	1–7	0.74	−0.565	−0.296
Identified regulation	5.63	1.16	1–7	0.70	−1.105	1.372
Autonomy	3.50	0.86	1–5	0.70	−0.374	−0.237
Competence	3.95	0.79	1–5	0.76	−1.035	1.193
Relatedness	4.27	0.72	1–5	0.71	−1.358	1.779
Center climate	4.03	0.74	1–5	0.85	−0.778	−0.021
Teacher climate	4.20	0.67	1–5	0.69	−1.070	1.356
Prosocial behavior	4.11	0.71	1–5	0.89	−0.358	−0.094
Antisocial behavior	2.28	0.80	1–5	0.87	1.270	1.634
Social responsibility	5.26	0.75	1–6	0.86	−1.991	5.728
Personal responsibility	5.19	0.84	1–6	0.82	−1.908	4.823
Violence	2.01	0.73	1–5	0.95	0.776	−0.187

Note: M = mean; SD = standard deviation; α = Cronbach’s value.

**Table 2 behavsci-11-00036-t002:** Correlations between variables and age.

Variables	1	2	3	4	5	6	7	8
Age (1)	1	−0.382 **	−0.588 **	−0.320 **	−0.293 **	0.009	0.348 **	0.542 **
Responsibility (2)		1	0.629 **	0.494 **	0.648 **	0.321 **	−0.232 **	−0.261 **
School Climate (3)			1	0.489 **	0.673 **	0.148 **	−0.232 **	−0.476 **
PMI (4)				1	0.582 **	0.266 **	−0.065	−0.238 **
Autonomous motivation (5)					1	0.228 **	−0.030	−0.229 **
Prosocial behavior (6)						1	0.107 *	0.107 *
Antisocial behavior (7)							1	0.513 **
Violence (8)								1

Note: PMI = psychological mediator index; * *p* < 0.05; ** *p* < 0.01.

**Table 3 behavsci-11-00036-t003:** Multivariate analysis according gender.

	Boys		Girls			
	M	SD	M	SD	F	*p*
Responsibility	5.12	0.79	5.35	0.61	10.177	0.002 **
School climate	4.02	0.66	4.21	0.63	9.734	0.002 **
PMI	3.87	0.68	3.95	0.59	1.814	0.179
Autonomous motivation	5.44	1.10	5.60	0.91	2.840	0.093
Prosocial behavior	4.00	0.72	4.23	0.69	10.979	<0.001 **
Antisocial behavior	2.48	0.83	2.07	0.71	29.662	<0.001 **
Violence	2.09	0.73	1.94	0.73	5.254	0.022 *

Lamda de Wilks (λ) = 0.893 (f = 7.206) * *p* < 0.05; ** *p* < 0.01.

**Table 4 behavsci-11-00036-t004:** Correlations between variables.

Variables	1	2	3	4	5	6	7
Responsibility (1)	1	0.629 **	0.648 **	0.494 **	0.321 **	−0.232 **	−0.261 **
School climate (2)		1	0.673 **	0.489 **	0.148 **	−0.232 **	−0.476 **
PMI (3)			1	0.582 **	0.228 **	−0.030	−0.229 **
Autonomous motivation (4)				1	0.266 **	−0.065	−0.238 **
Prosocial behavior (5)					1	0.107 *	0.107 *
Antisocial behavior (6)						1	0.513 **
Violence (7)							1
CR	0.771	0.785	0.751	0.804	0.811	0.880	0.952
AVE	0.280	0.646	0.508	0.511	0.685	0.786	0.909

Note: PMI = psychological mediator index; CR = composite reliability; AVE = average variance extracted; * *p* < 0.05; ** *p* < 0.01.

## Data Availability

https://osf.io/5ydxv/?view_only=a0f8c9bef4ed4c4bbf3beb2949099022.

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
