# Peer review of "School Climate and Responsibility as Predictors of Antisocial and Prosocial Behaviors and Violence: A Study towards Self-Determination Theory"

_behavsci, 2021, doi:10.3390/bs11030036_

Round 1

Reviewer 1 Report

This study examines whether contextual factors (school climate) and personal factors (responsibility) can prevent the adoption of antisocial behavior and violence, and encourage prosocial behavior. Through a SEM model, motivation and basic psychological needs are considered as underlying mechanisms in this relation. Although this research focuses on relevant research questions and although it is particularly interesting that the entire sequence is investigated at once, I have some concerns that I would like to address. 

Title

The authors would do better to mention that this research is conducted within the school context (e.g. by replacing “social” by “school”, or by adding “at school” to the first sentence.

Abstract

From the abstract, it cannot be understood that a SEM model will be tested, with motivation as an underlying mechanism between the needs and the anti/prosocial behavior (i.e. “the needs can improve the autonomous motivation and finally have positive consequences…” would be better replaced with “the needs can improve autonomous motivation, which in turn...”).

Introduction

The introduction is a loose sequence of research that has already been conducted on the same (or related) variables. Here is a suggestion to re-structure the introduction (in my opinion) more logically:

  • 1) Discuss why it is relevant to investigate pro/antisocial behavior and violence within the school context.
  • 2) Discuss how the school climate might influence these behaviors (1st antecedent). The sentence “In this way, several recent research have already stated that a low level of social skills…” is mentioned too early in the current introduction, and is better dealt with in the next paragraph. The paragraph on school context can be extended to what is meant by “school climate” (some sort of definition) and how this relates to other constructs in previous research. On the other hand, it’s better to save the link with basic psychological needs and motivation for the 4th section.
  • 3) Discuss how responsibility might influence these behaviors (2nd antecedent). In this current section, it is not clear why a fifth domain is mentioned, if this additional domain is not taken into account in the next sentence (and further).
  • 4) Discuss how the basic psychological needs and subsequent motivation may explain these links (as an underlying mechanism between the antecedents and the outcomes). In addition, the Hierarchical model of Vallerand is better discussed explicitly (what does this sequence look like?), instead of just mentioning it.

Instruments

  • L100 and L119: There is mention of a pre-test, despite the cross-sectional nature of the study?
  • PSRQ: The two types of responsibilities are taken together in the analyses, so it would be better to also mention the reliability of this composite scale.
  • CECSCE: Please clarify that center = class and that school = class + teacher?
  • EME: It is confusing that there is mention of "external causes and amotivation", as well as "seven subscales" while only autonomous motivation is assessed. For this reason, it is not clear with how many items autonomous motivation (and each subscale) is assessed.
  • CUVE: The number of items (per subscale) and the reliability per subscale are missing. For other scales, information about the subscales is always given, even if they are eventually taken together. It is best to be consistent and do the same here.
  • TISS: The information regarding the number of items is missing.

Statistical analyses

  • P5: In the plan of analysis, it is announced that the NFI and SRMR will be used to analyze the model fit, but this turns out not to be the case later on.
  • P5-6: It is not clear how the background variables are dealt with in the main analyses? Please clarify this. It would be interesting to see how the sample characteristics are related to the study variables (e.g. conducting a MANOVA for gender, including age in the correlation table).
  • P6: The RMSEA does not meet the cut-off value, and yet it is called a good model fit.
  • P7: In the figure, the outcome-variables refer to a first and second variable (e.g. Antisocial1 and Antisocial2). Do these numbers represent primary and secondary school respectively? Please clarify. However, to examine the differences between primary and secondary schools, a multigroup SEM analysis is needed.
  • To conclude that school climate and responsibility lead to positive behavioral outcomes, indirect (and preferably direct) effects must be reported.
  • Certain studies mentioned in the discussion section should already be dealt with in the introduction section (e.g. 43, 52), because they are so closely related to the current research questions.

General

As a general point, it would be beneficial to proof read the article very carefully. There are grammar errors and many sentences require rephrasing.

Reviewer 2 Report

The English expression in this article is problematic and it is serving to obfuscate some of the author's points. I recommend thorough proofreading by a professional operating at native English level, then I will be happy to re-review. This article is not yet ready for publication due to this factor, which also makes it very difficult to review properly. 

For your reference, here are two verbatim examples in the abstract that show the authors are not yet able to express their study clearly enough in English: 

The main results saw that the most part of
18 variables were correlated positive and directly except with antisocial behaviors and violence.

In conclusion, school climate and responsibility can improve positive consequences to the classroom, specifically, prosocial behaviors and reducing violence and antisocial behaviors.

Reviewer 3 Report

Social Climate and Responsibility as predictors of antisocial and prosocial behaviors and violence. A study towards the Self-determination theory

02/02/2021

The present study tries to analyze the relationship through a SEM model of different variables and psychosocial factors. I consider that the work deals with a relevant topic, although I consider that there are some points that question the conclusions reached by the authors. With the intention of being contractionary in my work as a reviewer, I present some points that I consider could be improved:

  • What is the empirical background of the Personal and Social Responsibility Model (TPSR)? Have you been empirically tested before? What results has it shown? It talks about the results of a review of the literature on the model, but it is not clear what this review concludes, nor does it answer the questions I previously posed. Since this is the background of the work, it would need further development.
  • The study tries to establish relationships for the first time between different factors proposed by different theories, but why? That is, what leads the authors to propose a joint study of these factors? This would need more theoretical support.
  • Is the chosen sample representative of the population to be analyzed? What criteria was established when choosing the sample size? Could the authors provide information on the statistical power for that sample size? This would help to know if the sample is representative of the population.
  • The internal consistency or reliability of the scales used is provided, but are there data on other psychometric characteristics (eg, convergent validity or divergent validity) that help to understand the suitability of these scales?
  • The study has underage students as a sample. Does the study have the ethical support of an evaluation committee?
  • As estimator for factor analysis they used ML. This estimator is ideal for continuous data that follows a multivariate normal distribution. Was the data created for data distribution and assumptions for factorial treatment?
  • The work uses different scales that measure different latent variables, but has the validity of the scales been analyzed or are there precedents in the chosen sample?
  • The SEM model shows a somewhat poor fit (CFI = .894). It is possible that there are measurement errors that their correction or adjustment will help to improve the fit, provided they have a theoretical justification.
  • The work does not present limitations and prospective.
  • The whole paper needs a good copy edit

Round 2

Reviewer 1 Report

I am happy with the authors' responses to my comments and feel that the revised manuscript has been improved as a result. Nevertheless, I believe that to properly answer the research questions, it is necessary to report the direct and indirect effects of the Structural Equation Modeling

Author Response

Dear reviewer,

You were totally right. We did not answer this question in the last round. Two new sentences have been added in 3.3. Structural model section.  One for the direct effect and a second one for the indirect effect. Please, do not hesitate if you think it is necessary more requirements.

Your sincerely,

Alfonso Valero-Valenzuela. 

Reviewer 3 Report

I am happy to see how the work has improved substantially throughout the review process. The authors have given a satisfactory answer to all the questions that posed a problem for the manuscript.
I consider that the work is of sufficient quality to be accepted.
Best regards.

Author Response

Dear reviewer,

Thank you very much for giving us the opportunity to learn with your support. 

Best wishes!

Alfonso Valero-Valenzuela.